# Gaelic4Girls—The Effectiveness of a 10-Week Multicomponent Community Sports-Based Physical Activity Intervention for 8 to 12-Year-Old Girls

**DOI:** 10.3390/ijerph17186928

**Published:** 2020-09-22

**Authors:** Orlagh Farmer, Kevin Cahill, Wesley O’Brien

**Affiliations:** 1Sports Studies and Physical Education Program, School of Education, 2 Lucan Place, Western Road, 0000 Cork, Ireland; Wesley.obrien@ucc.ie; 2School of Education, Postgraduate Diploma in Special Educational Needs, University College Cork, 0000 Cork, Ireland; K.Cahill@ucc.ie

**Keywords:** physical activity, fundamental movement skills, psychological correlates of physical activity, pre-adolescent girls, organized youth sport, multi-component intervention

## Abstract

Girls are less active than boys throughout childhood and adolescence, with limited research focusing on female community sports-based programs. This study aims to assess the effectiveness of a multi-component, community sports-based intervention for increasing girl’s physical activity (PA) levels, fundamental movement skill (FMS) proficiency, and psychological wellbeing, as relative to a second treatment group (the traditionally delivered national comparative program), and a third control group. One hundred and twenty female-only participants (mean age = 10.75 ± 1.44 years), aged 8 to 12 years old from three Ladies Gaelic Football (LGF) community sports clubs (rural and suburban) were allocated to one of three conditions: (1) Intervention Group 1 (n = 43) received a novel, specifically tailored, research-informed Gaelic4Girls (G4G) intervention; (2) Intervention Group 2 (n = 44) used the traditionally delivered, national G4G program, as run by the Ladies Gaelic Football (LGF) Association of Ireland; and (3) Control Group 3 (n = 33) received no G4G intervention (group 1 or 2) conditions and were expected to carry out their usual LGF community sports activities. Primary outcome measurements (at both pre- and 10-week follow up) examining the effectiveness of the G4G intervention included (1) PA, (2) FMS and (3) Psychological correlates (enjoyment levels, self-efficacy, peer and parental support). Following a two (pre to post) by three (intervention group 1, intervention group 2, and control group 3) mixed-model ANOVA, it was highlighted that intervention group 1 significantly increased in PA (*p* = 0.003), FMS proficiency (*p* = 0.005) and several psychological correlates of PA (*p* ≤ 0.005). The findings demonstrate that the 10-week, specifically tailored, research-informed G4G intervention is a feasible and efficacious program, leading to a positive effect on the physical and psychological wellbeing of pre-adolescent Irish girls, relative to the traditionally delivered national G4G comparative program and control group conditions.

## 1. Introduction

It is widely reported that girls are less physically active than boys throughout childhood [1], and the age-related decline in physical activity (PA) participation, particularly from early adolescence onwards, is steeper for girls when compared to boys [2]. Pre-adolescence (11–12 years of age) is a critical period of change in the PA participation levels of girls [3], and finding ways to help girls to become more physically active at this age is important for short and long-term health. The gender disparity in PA participation has highlighted the need to develop and evaluate interventions, specifically targeting at-risk youth [4].

Club-based participation in organized youth sport (OYS) during childhood and adolescence contributes considerably to leisure-time PA for health-enhancing benefits [5] and has the potential to increase overall PA levels in young people [6,7], specifically among girls [8]. OYS participation is also associated with important psychological benefits, which include increased wellbeing, self-efficacy motivation, perceived competence and confidence, attitude and enjoyment of PA, as well as external socioecological factors such as positive family, peer and coach support [9,10,11,12,13]. Studies have examined both peer and family support for PA, and results have shown that some studies have found a positive association [14,15], while others have not [16]. Furthermore, according to socio-ecological models [17,18], these associations between peer and parental factors and adolescent girls’ PA could be mediated by personal factors, suggesting that peer and parental factors may influence girls’ PA participation indirectly. This implies that these personal factors are also associated with adolescent girls’ PA, specifically self-efficacy [19,20]. In regard to specific PA programs for female youth, it is recommended to incorporate psychological skill instructions that aim to enhance self-esteem, positive body image, positive attitudes towards PA, and motivation to participate in PA [21,22,23]. Programs should aim to also provide social environments that offer a variety of activities that girls may consider enjoyable [24,25].

Evidence also highlights that a positive relationship exists between participation in OYS and fundamental movement skill (FMS) proficiency in children [26,27], allowing children to cumulatively acquire transferable movement skills that give them a sense of movement competence and confidence [28]. A Portuguese cross-sectional study (n = 973) reported that primary school children participating in OYS are less sedentary and participate in more frequent bouts of moderate to vigorous physical activity (MVPA) [8], when compared to those who do not participate in organized sports. Although OYS does not necessarily prevent the decline in PA participation during adolescence [29], research does continue to illustrate that children who maintain their involvement in sport are also more likely to participate in PA during adolescence [30,31] and into adulthood [31,32].

Encouragingly, sport occupies a prominent place in Ireland’s national cultural identity, which is reflected in the high prevalence (58–80%) of 10 to 18-year olds who participate in some form of OYS, on at least one occasion in the week [33]. The Global Advocacy for Physical Activity (GAPA) has identified participation in sports as one of the seven worldwide “investments that work” for improving youth PA levels [34], and for these reasons, community sports-based settings are recognized as key health settings in promoting knowledge of PA [35] and healthy lifestyle behaviours [36]. These youth sport participation outcomes have obvious short-term benefits; however, this domain may also help to develop “physical literacy” (PL), and thereby support the continuum of lifelong PA. Based on Tremblay et al.’s work [37], PL represents the successful interaction of four inter-related core domains: (a) physical fitness (cardiovascular fitness, muscular strength and endurance, flexibility, and coordination); (b) fundamental motor skills (e.g., catching and throwing a ball); (c) PA behaviours; and (d) psychological/cognitive factors (attitudes, knowledge, and feelings). Furthermore, it is important to recognize that the value of the PL domains provides a powerful lens for PA, in relation to motor skill outcomes, environmental context, and broader psychological outcomes, including social and affective learning processes [38].

Despite the widely endorsed benefits for OYS, there are a lack of effective intervention strategies to promote sustainable PA participation in female youth [39], particularly within a sporting context [40]. In the last decade, there has been a gradual increase in the number of randomized controlled trials (RCTs) that have evaluated the impact of PA-based interventions on young girls [41,42,43]; however, minimal evaluations of pre-adolescent girls within a community sports-based setting exist. Of those RCTs which have been undertaken, results have found small effects when PA was measured objectively [44,45]. The need to promote higher habitual MVPA engagement amongst children (particularly girls) in the youth sport context has been underlined as a prudent strategy [46], particularly as knowledge translation from youth PA research to practice in the community is weak [47,48]. Furthermore, it is not yet fully understood how OYS in female youth could be optimized to facilitate continued participation and increased PA, specifically in a female sport context, such as Ladies Gaelic Football (LGF) (a team field sport that is popular in Ireland and organized through the Ladies Gaelic Football Association). The Gaelic4Girls (G4G) research project is part of an existing national LGF program in Ireland for 8–12-year-old female adolescents [49].

The primary aim of this three-armed non-randomized controlled trial (NRCT) was to assess whether 8- to 12-year-old child and pre-adolescent girls who attended a multi-component research-informed tailored 10-week G4G community sports-based intervention could (i) increase their overall PA levels, (ii) FMS proficiency, and (iii) psychological wellbeing when compared to an intervention group 2 and a control group 3. The current paper presents, as far as we are aware, the first NRCT conducted in an Irish ‘Gaelic’ Games OYS context. It is hypothesized that intervention group 1 would demonstrate significant improvements in all primary outcome variables when compared to those within the intervention group 2 and control group 3 conditions.

## 2. Materials and Methods

### 2.1. Study Design

This was a quasi-experimental, non-randomized controlled before-and-after design, with a mixed-methods approach. This study adopted a three-arm NRCT design, with a specifically tailored, research-informed G4G intervention group (1), a second treatment intervention group (2) and control group (3). The study followed the transparent reporting of evaluations with nonrandomized designs (TREND) statement for reporting [50]. One hundred and twenty female-only participants (M age = 10.75 ± 1.44 years), aged 8 to 12 years old from three LGF community sports clubs (rural and suburban) in Ireland were recruited to partake in one of three study conditions: intervention group 1 (n = 44), the club that received the research-led G4G intervention; intervention group 2 (n = 43), the club that used the existing ‘traditional’ national G4G program structure; and control group 3 (n = 33), the club that received no G4G intervention condition, and was expected to carry out their usual LGF community sports activities. Assessment measures were conducted at pre-intervention (March 2018) and were repeated at post-intervention (June 2018) phases. All subjects gave their informed consent for inclusion before they participated in the study. The study was conducted in accordance with the Declaration of Helsinki, and the protocol (2015-005) was approved by the Social Research Ethics Committee of the researchers’ institution (University College Cork) in 2016.

### 2.2. Recruitment, Participants and Setting

Convenience sampling was used to recruit participants from the 18 girls’ only primary schools (rural and urban), from three local LGF clubs in the Cork and Kerry regions (Munster Province, Ireland). All schools were in close proximity to the local, designated LGF club and research team. To be considered eligible for this study, participants (girls only) were formally enrolled in between the years of second to sixth class of primary school (aged 8 to 12 years old) and had expressed an intention to attend the program for its full duration. Informed parental consent and child assent were the requirements for eligible participation in this study. Information sheets and consent forms were administered to students who expressed an interest in taking part in G4G. Informed assent for participation was granted by all participants, and written consent was obtained from their parent(s)/guardian(s), prior to the physical and psychological data collection measurements. For coaches to be eligible to partake in the program, recognized coaching credentials from the national LGFA, and previous experience of coaching girls within the existing community sports-based setting was required; all participants were free to withdraw from the research at any stage.

Prior to the commencement of the research-informed G4G intervention, the leading researcher emailed the school principal from the five surrounding primary schools, calling for expressions of interest for the lead researcher to visit the school and inform the girls about the G4G program (intervention group 1). Subsequent to the granted approval from the five school Principals, the lead researcher, along with the two G4G coordinators (head club coaches from the selected LGF intervention club) visited the five primary schools, where a full outline of the G4G intervention and the associated data collection measurements were provided as part of the intervention group 1 condition. Club coordinators from intervention group 2 and control group 3 locations were also contacted by the lead researcher via phone call, and a full brief of the associated data collection measurements were again outlined. One hundred and sixty-five participants were invited to participate in this study, with a total of 137 providing full consent (intervention group 1: n = 50; intervention group 2: n = 49; control group 3: n = 38). In total, 120 participants had fully available data at both pre- and post-data collection time points (intervention group 1: n = 44; intervention group 2: n = 43: control group 3: n = 33).

#### 2.2.1. Gaelic4Girls Research-Informed and Tailored Intervention Group 1

The G4G intervention, underpinned by the self-determination theory (SDT) [51], and elements of the social-ecological model (SEM) [52] (see Table 1) is a multi-component PA and FMS community sports-based intervention. Participants selected for the research informed G4G program received the 10-week intervention, consisting of 1 × 60 min, specifically tailored LGF session per week (10 sessions in total between March and May 2018). This G4G intervention includes three major components, as guided by the associated theoretical constructs: the (1) participant component, (2) coach education component, and (3) parent/guardian/community component.

#### 2.2.2. Gaelic4Girls Nationally Delivered and Existing G4G Intervention Group 2

Participants allocated to intervention group 2 (n = 43) received the 10-week ‘traditional’ existing G4G program, as run by the LGFA. In comparison to intervention group 1, this program consisted of 1 × 60-min introductory LGF sessions per week (10 sessions in total between March and May 2017), without research-informed content, coach education professional development workshops, or any additional digital or hardcopy resources. Similar, to intervention group 1, the ‘traditional’ G4G program included three major components: (1) participant component, (2) coach education component, and (3) parent/guardian/community component.

#### 2.2.3. Control Group 3

Participants selected for control group 3 received no G4G program, resources or coach education workshops. The participants trained as they would normally in their local LGF club setting over a 10-week period.

### 2.3. Outcomes

#### 2.3.1. Data Collection

Pre- and post-intervention assessments were conducted by trained research assistants at the community sports club and local primary school settings. Pre-testing took place on week 1 of the study, while post-testing took place upon completion of week 10 of the study. To enhance the quality of the data across all collection sites, the research assistants (all pre-service physical education teachers) were formally trained in the standardised measurement procedures and protocols associated with FMS and self-report questionnaires. Specifically, each research assistant attended a 3-h robust field training workshop and was given detailed manuals, checklists and scripts to read, when informing the participants about the measures.

Objective measurements, such as FMS were conducted in the community sports club setting hall, with a ratio of one researcher to five child participants (1:5). Subjective pre- and post-self-report PA measurements took place in a supervised classroom or computer lab, and the ratio of a researcher to child participants was 1:10. All questionnaires were completed online through the tool “Survey Monkey”.

#### 2.3.2. Physical Activity Self-Report Assessment

Moderate to vigorous PA (MVPA) was assessed using a modified version of the Physical Activity Questionnaire for Older Children (PAQ-C) [56]. Studies have established the reliability and validity of the 7-day recall of children [57]. The PAQ-C for this study included 15 physical activities, 10 leisure/free-time activities, activities in school (Physical Education), transport activities (walking to and from school), and other activities. The participants were told to recall what activities they had engaged in the previous seven days and how many times and the number of minutes they participated in each of these activities for. Habitual PA was also assessed using two questions from the Physician-based Assessment and Counseling for Exercise (PACE) questionnaire: how many days in the last week (PACE 1) and a usual week (PACE 2) does the subject do at least 60 min of PA.

#### 2.3.3. Fundamental Movement Skills

The FMS proficiency of participants across seven-movement skills was assessed in conjunction with the behavioural components from three established instruments, namely the Test of Gross Motor Development (TGMD) [58], Test of Gross Motor Development-2 (TGMD-2) [59], and the Get Skilled Get Active resource [60]. Each of these instruments and their associated protocols has established validity and reliability in children and are designed to give an objective measurement of gross motor skill proficiency. The selected seven FMS included culturally relevant skills to LGF: three locomotor skills (run, skip, and vertical jump), one stability (balance) skill, and three object control skills (stationary dribble, catch, kick), which combine to give an overall maximum raw score of 58.

Participants performed the skill on three occasions, including one familiarisation practice, and two performance trials, as reported in previous Irish movement skill data assessments [61]. Participant performance, along with the execution of the required skill, were recorded using digital video cameras (2 × Canon type Legria FS21 cameras; Canon Inc., Tokyo, Japan) and 2 × Apple iPads to allow for greater measurement scrutiny and accuracy of measurement precision during analysis. Once data collection was completed, the principal investigators were required to reach a minimum of 95% inter-observer agreement for scoring all seven FMS.

#### 2.3.4. Psychological Correlates of PA

A variety of psychological outcomes were assessed using existing questionnaires that have demonstrated reliability and validity for use with this age group, namely, self-efficacy (SE), perceived physical self-confidence (PSC), PA enjoyment, PA attitudes, perceived peer and parental social support.

Participants’ perceived (PSC) levels were assessed using the PSC scale [34], which has shown excellent test–retest reliability, and internal consistency, with a Cronbach alpha coefficient of 0.94 [34]. In the current study, the Cronbach alpha was 0.88, suggesting very good internal consistency for the perceived PSC scale. The PSC scale consists of 15 questions in which participants rate their confidence at performing 15 separate FMS. The identified 7 FMS included within this instrument are considered central to the Irish youth sporting culture [8,33]. Participants rated their confidence at performing each skill on a Likert scale of 1–10, with “1” being not confident at all, and “10” being very confident. This present study assessed 7 of the 15 FMS from the PSC scale, consistent with the 7 actual movement skills assessed. The maximum perceived PSC score which could be achieved was 70 (if participants scored their confidence at 10/10 for performing all 7 skills).

Participants’ SE was assessed using a modified version of the Children’s Physical Activity Self-Efficacy Survey, an 11-item scale modified by Sherwood et al. [62]. Examples of the statements included “I could do PA even if I was tired”, and “I could do PA even if I had to exercise on my own”. Participants were asked to indicate how true each statement was to them (“very true” to “not very true at all”, range 1–4). Participants’ PA enjoyment was assessed using a modified version of the Choices questionnaire [63]. The scale for this study consists of ten statements (e.g., “I enjoy it”, “I feel good”). Participants were asked to indicate their agreement level of each statement (“agree a lot”, to “disagree a lot”, range 1–5), and the participants’ selected the statement that best described them. PA attitudes were assessed using the scale from the Rhodes and Smiths [64] questionnaire, consisting of 14 statements (e.g., “doing PA everyday would be important for me”, “I could do PA every day if I wanted to”). Similar to the PA enjoyment scale, participants were asked to indicate their agreement level of each statement (“agree a lot”, to “disagree a lot”, range 1–4), and the participant selected the statement that best described them.

Participants perceived peer/social support was assessed using the Birnbaum et al. [65] modified version of the Choices questionnaire, which consists of five statements (e.g., “Do your friends encourage you to do physical activities and play sports?”, “Do friends tell you that your doing well in physical activities or sports?” etc.). Participants were asked to indicate how often peer support was provided for each of the statements (“none” to “everyday”, range 1–5). Participants perceived family support was assessed using the Saunders et al. [66] modified version of the Choices questionnaire, which consists of five statements. Participants were asked to indicate how often (“none” to “everyday”, range 1–5) someone in their house/member of family “encouraged them to do physical activities or play sport”, or “provided transport (e.g., a spin) to a place where you can do PA and sport”.

With regards to the aforementioned psychological variables, the more positive the statement in the questionnaires, the higher the values, with negatively worded statements reversed and recoded. In the current study, the Cronbach alpha coefficient for all psychological measures were calculated, and all scales have acceptable internal consistency. These internal consistency values were greater than 0.70, in all cases representing acceptable internal consistency values [67].

#### 2.3.5. Data Analysis

The combined PA, FMS and psychological correlate dataset was analysed using SPSS version 20.5 for Windows (IBM, Armonk, NY, USA). Participants with incomplete data for a given variable were excluded from the analysis. Specific percentages of missing data for each of the variables are as follows: PA (2.5%), LOM (1.7%), OC (1.7%), SE (1.7%), PSC (5%), PA attitudes (1.7%), peer support (4.2%), and familial support (5.8%). Descriptive statistics and frequencies for all PA, FMS and psychological variables were employed to describe participants’ characteristics. Comparability of the three groups at pre-test was ascertained using an independent sample t-test. The pre- to post-test variable changes was evaluated using repeated-measures analysis of variance (ANOVA) to determine time and time-by-group differences. All statistical assumptions were tested for normality, outliers, and homogeneity of variance. Specifically, a two (pre to post) by three (intervention group 1, intervention group 2 and control group 3) mixed model analyses of variance (i.e., between-within subjects) was conducted to analyse any main effects and time x group interactions for PA, FMS proficiency, and the psychological variables. Levene’s test was used to determine whether variances were equal between each group.

To reduce the risk of Type I error, a Bonferroni adjustment to the alpha level was made by dividing *p* < 0.05 by 10 (number of all variable comparisons); thus, the alpha level was set to *p* < 0.005. In instances where significant interaction effects were found, post hoc comparisons were carried out using the Tukey honest significance difference (HSD) to determine where the differences occurred. Effect sizes of significant differences were evaluated using partial eta-squared (η2). To calculate effect sizes, Cohen d values [67] were applied, with d = 0.2 representing a small effect size, 0.5 representing a medium effect size, and 0.8 representing a large effect size.

## 3. Results

### 3.1. Descriptive Statistics

All three intervention and control groups (N = 120) were similar for most pre-test characteristics, with a one-way ANOVA revealing no significant differences between groups for any of the primary outcome variables. Participants mean scores for all primary outcome variables pre and post-test (PA, FMS and psychological), as split by group condition are shown in Table 2.

### 3.2. Physical Activity

A mixed between-within subject’s analysis of variance was conducted to assess the impact of the three different groups (intervention group 1, intervention group 2, and control group 3) on participants’ self-reported PA, across two time periods (pre- and post-test). There was a significant interaction effect between program type (group) and time for PA, Wilks’ Lambda = 0.83, F (2, 115) = 11.77, *p* < 0.0001, partial eta squared = 0.170. Post-hoc tests (adjusted Bonferroni) revealed that there was a significant PA difference between intervention group 1 (mean change = 39.7, SD = 81.66, *p* = 0.003) and intervention group 2 (Mean change = −6.13, SD = 76.45, *p* < 0.002), from pre- to post-time periods. Similarly, there was a significant PA difference between intervention group 1 (mean change = 39.7, SD = 81.66, *p* = 0.003) and control group 3 (Mean change = −2.1, SD = 5.18, *p* = 0.004), from pre- to post- time periods. Conversely, there were no significant PA differences between intervention group 2 and control group 3 from pre- to post-periods. In terms of the direction of the results, the findings suggested a small PA difference (Cohen d = 0.03) between the three groups; only intervention group 1 showed a PA increase from pre- to post-periods. For intervention group 2 and control group 3, however, a reduction in PA was observed across the two time periods. Mean PA differences from pre- to post- in terms of self-reported weekly moderate-to-vigorous physical activity (MVPA) minutes among the groups (n = 120) can be seen in Figure 1.

### 3.3. Fundamental Movement Skills

Following the two by three mixed-model ANOVA, there were significant increases in overall FMS proficiency, from pre- to post-time periods for those in intervention group 1 when compared to intervention group 2 and control group 3 (*p* = 0.003) (see Figure 2). Specifically, results showed a small- to-moderate significant main effect for the three groups (F (2,115) = 6.16, *p* < 0.003, η^2^ = 0.09). There was a significant interaction effect between program type (group) and time, Wilks’ Lambda = 0.68, F (2,115) = 27.71, *p* < 0.0001, η^2^ = 0.325. Follow-up post-hoc indicated that participants in intervention group 1 (mean change = 1.86, SD = 4.78) reported significantly greater increases (*p* = 0.005) in overall FMS proficiency, when compared to those in control group 3 (Mean change = −1.21, SD = 4.14) only.

For locomotor skills, when comparing the three program (group) types (between-subjects effect), there was a significant interaction effect observed (F (2,115) = 21.98, *p* = 0.000, partial eta squared = 0.28), suggesting small-to-moderate differences only. Post-hoc tests revealed (adjusted Bonferroni) that only intervention group 1 showed a small increase in locomotor skill proficiency over time. Both intervention group 2 and control group 3 showed a small reduction in locomotor skill proficiency over time, as shown in Figure 2.

For object control skills, when comparing the three types of intervention/control groups (between-subjects effect), there was a small-to-moderate main effect observed (F (2,115) = 3.46, *p* = 0.035, partial eta squared = 0.057). There was a small but non-significant (*p* = 0.060) increase in object control proficiency scores for participants in both intervention group 1 (mean change = 0.55) and intervention group 2 (mean change = 0.02). Similar to locomotor skills, a small but non-significant (*p* = 0.100) reduction in scores for those in control group 3 was shown (mean change = 0.90).

### 3.4. Psychological Correlates of PA

There was no significant main effect for the three groups across all psychological variables (see Table 3). However, there were significant interaction effects for time within and between groups across all the psychological variables. Follow-up post-hoc tests revealed statistically significant differences for SE, PA enjoyment, and attitudes towards PA between the three groups, particularly for intervention group 1, when compared to their control group 3 counterparts. Post-hoc comparisons (adjusted Bonferroni) revealed statistically significant differences in mean scores between the three groups, with small-to-moderate effect sizes observed (using Cohen’s d value in Table 3). Table 3 presents the statistically significant differences for all significant psychological correlates of PA.

## 4. Discussion

The main purpose of the G4G NRCT was to assess the effectiveness of a multi-component, community sports-based PA intervention for increasing girls’ PA levels, FMS proficiency, and psychological wellbeing, when compared to an intervention group 2 and a control group 3. It is hypothesized that the intervention group would demonstrate significant improvements in all primary outcome measures, in excess of those within the comparison and control groups.

The PA data showed small, but significant increases in weekly minutes of self-reported MVPA, from pre- to post-time periods for those in intervention group 1, unlike those in intervention group 2 or control group 3. This positive PA observation may be due to the fact that the coaches in the intervention group one were guided by research-informed implementation coaching principles, and were exposed to evidence-based frameworks, which were designed to guide the planning, delivery, and evaluation of the organized weekly G4G sessions in the community sports-based club. Coaches in the intervention group one, therefore, had specific pedagogical guidance which sought to influence participants’ PA levels, alongside focusing an increase in PA intensity during the weekly coaching sessions [68]. The SHARP pedagogical principles [54] have recently proved successful in supporting teachers to increase 7- to 9-year-old children’s MVPA by 30%, in primary PE lessons.

Additionally, it is plausible that the increase in self-reported minutes of MVPA among intervention participants in group 1 may be due to the Teaching Games for Understanding (TGfU) [53] nature of the G4G station-based sessions. Participants in the intervention group were exposed to a variety of FMS, and LGF sport-specific skills, through a TGFU instructional model [53], within a rotatory station-based approach. Each 10-min station allocated a significant amount of time to learning the skills of LGF via modified and small-sided games, which directly reinforced heightened periods for PA and minimising opportunities for physical inactivity among participants [69]. Previous research studies utilizing the TGfU approach have proven successful in terms of increasing PA among children and adolescents in both school and community sports-based contexts [70,71]. Other pedagogical instructional models in teaching and coaching can be effective in increasing the short- [72,73,74] and long-term [75,76] PA levels of children and adolescents. Previous pedagogical strategy interventions have produced a mean PA increase of 6.27% in children’s MVPA during physical education lessons, with intervention groups spending 14% more time in the desired PA intensity when compared to control groups [77].

The present results revealed that youth in the intervention group also reported significantly greater increased scores in their overall FMS proficiency over time when compared to both groups. Significant increases in participants overall locomotor skills proficiency was evident among intervention group 1 from pre- to post-intervention, in comparison to the two other groups. No significant differences were found in the object-control skill subset. Specific training of coaches is assumed to play a role in the improvements noted in the research informed intervention group. Increases in the intervention group’s overall FMS proficiency may also have been due to the fact that children received performance-related feedback on FMS and sport-specific skills. In addition to this, the inclusion of the G4G FMS dance provided an additional opportunity for participants in intervention group 1 to practice basic FMS (e.g., coordination, skipping), combined with the sport specific LGF skills into a fun dance sequence. Creative dance is a pedagogical approach of exploration and experimentation, it teaches children to use their creative abilities that lead them effortlessly in the development of movement abilities, and it also increases the availability and willingness of children to engage in motor activities [78]. Among youth, dance programs have facilitated changes in physical outcomes, such as fitness (e.g., [43,79]), as well as in psychological outcomes (e.g., [79,80,81]). In the present study, the dosage of FMS specific time for the intervention group participants (including the weekly 10-min FMS station (100 min) and FMS dance practice and performance (300 min)) totalled 400 min throughout the 10-week intervention duration. Therefore, the research-informed intervention group 1 were exposed to a higher dosage of FMS activities, and it seems likely that they would improve their overall FMS proficiency to a higher level than those in intervention group 2 or control group 3.

Recent findings from the Vallence et al. [82] CHAMPS Study-DK complement the current study’s findings, showing that weekly participation in organised sport over a 30 month period from 1067 Danish school-aged children (6 to 12 years) is positively associated with motor performance across the coordination fitness spectrum. Further to this, Bryant et al. [83] found that a 1 day per week PE intervention, focusing on FMS, resulted in the improvement of motor skills in children (mean age ± SD = 8.3 ± 0.4 years) within the intervention group (n = 82) relative to children in the control group (n = 83). Most recently, Costello & Warne [84] showed that a four-week FMS intervention effectively improves motor skills in 8- to 10-year-old Irish primary school children (n = 100). Findings from previous community sports-based youth interventions provide evidence that FMS supporting strategies for coaches, including the implementation of a station-based format, and a TGfU pedagogical approach may be an effective way to improve (1) FMS competencies for children, (2) deepen children’s knowledge of game tactics, and (3) heighten overall enjoyment levels for children [83,85].

In order to support coaches throughout the intervention, every coach received a comprehensive G4G Resource Pack, which was aligned to the evidence informed SAAFE [55] and SHARP principles [54]. The initial 2-h coach education workshop session (professional development prior to the start of the G4G intervention) alongside the 8 × 1-h coach education sessions with the lead researcher may have had an impact. Specifically, these professional development opportunities for coaches targeted face-to-face engagement with the intervention groups coaches, whereby there were consistent visual demonstrations on the specific FMS performance criteria and coaching cues. Additionally, the G4G resource pack and electronic video resources may have assisted the coaches in understanding FMS, as these resources further reinforced the key coaching skill performance criteria. The provision of Continuous Professional Development (CPD), similar to the Robbins et al., [86] “Girls on the Move” intervention protocol was adhered to in terms of discussing issues, and reinforcing G4G policies and procedures, which may have been a contributing factor to the coaches understanding and delivery of FMS. A similar approach was utilised by Cohen et al. [87], with 460 children (54.1% girls; age 8.5 ± 0.6 years) in the Supporting Children’s Outcomes Using Rewards, Exercise and Skills (SCORES) intervention, where teachers learnt about FMS and were instructed to provide students with specific feedback [87]. The literature widely acknowledges that youth PA interventions through the provision of non-formal coach education, such as CPD opportunities (training, resource manual, and opportunities to work with coaches and instructors) can positively support program implementation and childhood engagement in PA [88,89,90].

The present results revealed that participants in the intervention group also reported small, but significantly greater scores in some of the psychological correlates (see Table 3), including their SE and perceived PSC over time when compared to other two groups. Coaches in the intervention group were encouraged to utilise purposeful praise, positive reinforcement and performance-related feedback for the duration of the 10 weeks, which was highlighted as part of the CPD offering. Bandura [91] posited that SE, as a product of a complex of self-persuasion, relies on the cognitive processing of diverse sources of efficacy information, including mastery experiences, vicarious experiences, and verbal persuasion. Performance-related feedback given by coaches has the potential to have a profound effect on players’ performance, as well as on their perception of the motivational climate [92]. It would seem likely that participants in the intervention group were more susceptible (as observed) to heightening their SE and perceived PSC levels, than those in the comparison and control groups. In the context of OYS, a consistently positive association between SE, perceived competence, and participation in sport by children and adolescents is evident [19,93,94,95]. Recent Irish research [96] among 860 children (M age: 10.9 ± 1.16) found that physical SE mediates the movement competence and PA relationship, with the entire model explaining approximately 10.3% of the variance of PA. Findings of the current study highlight the need for interventions to target and improve movement competence as a whole for children.

Significant increases in participants PA enjoyment and attitudes was evident among the intervention group from pre- to post-intervention, in comparison to the two other groups. One explanation for this may have been the integration of the weekly 10-min “Team Challenges” fun station. This intervention component promoted the sense of inclusiveness, friendship and belonging to participants, while at this station, coaches also provided feedback to each of the participants in a fun non-competitive environment. The need for relatedness (SDT) and feeling connected to others (peers/coaches) can help cultivate the development of intrinsic motivation [97] for increased PA participation [20,98,99]. Previous studies have acknowledged that the experiences of enjoyment are a critical factor in motivating young individuals to continue participating in PA [100,101]. Studies have found that those who enjoy the benefits of physical activities have a favourable and positive attitude toward PA participation [102]. This was evident in the current study, with intervention group 1 showing significant gains in PA attitudes in comparison to intervention group 2 and control group 3 conditions. Within the PA context, enjoyment represents a positive attitude toward PA practice [103] and constitutes one of the most important correlates for PA participation [104].

Significant increases in participants perceived peer and familial social support was evident among intervention group 1 from pre- to post-intervention, in comparison to the two other groups. Social support was a key component of the research-informed intervention group 1 for participants (via positive reinforcement and praise). Peer influence within the G4G research-informed intervention may have been influenced through (1) modelling of PA (Team Challenge station), and (2) co-participation (TGfU stations) with participants. Within the research informed G4G intervention group, parents undertook the following (i) co-participated in the LGF skills with their daughter (G4G Skill Cards), (ii) attended a “Parent’s Evening” educational workshop on week 4 (led by the lead researcher), and (iii) were provided with a communicative support structure platform (G4G Facebook page and WhatsApp group) for the duration of the 10-weeks. The co-participation (e.g., performing physical activities together) and modelling communicative support structures may have led to increased parental social support and involvement with their daughter’s engagement in the program. Studies have examined both friend and family support for PA, and results have shown that some studies have found a positive association [14,15,105,106], while others have not [16]. Results of the current study are in line with Seabra et al. [107] who examined correlates of PA in schoolchildren aged 8–10 years, and found that perceived acceptance by peers in sports, and parental encouragement was positively related to PA in girls (n = 683). Some recent evidence would suggest that social support is a potentially important mediator of increased PA in young people [108,109,110,111], and has been targeted by several intervention programs [112,113].

The majority of research conducted to-date has taken place in school settings, with little attention given to multi-component OYS interventions, incorporating simultaneous physical, psychological and socioecological factors in the development and reinforcement of healthy PA behaviours [114]. Considering the dearth of research in an Irish OYS setting, specifically multi-component interventions targeting child and pre-adolescent girls in the Gaelic Games context, this is an area that merits exploration.

Limitations of the current study include the relatively small sample size of the intervention club, along with the non-randomized controlled research design, which limits the generalisability of the findings. The novelty of participating in intervention groups 1 or 2 may have demonstrated a heightened positive attitude towards PA and its correlates. These intervention conditions may have increased the participant–coach engagement with the intervention and influenced the findings in unknown ways. The study design was limited to a subjective method for the assessment of participants PA; a future recommendation would be to use more objective measurements of PA, such as accelerometers, or wearable technologies.

A strength of this study is the enhanced ecological validity, stemming from the naturalistic community sports based G4G intervention setting. Finally, a further strength of the study is the use of simple, innovative and feasible intervention strategies, based on a sound conceptual framework, which can be adapted in other similar OYS settings.

## 5. Conclusions

The results of this study indicate that the 10-week G4G research-informed multi-component community sports-based intervention led to improvements in (i) self-reported PA levels, (ii) overall FMS proficiency, and (iii) several psychological correlates of PA, including perceived SE, perceived PSC, PA enjoyment, attitudes toward PA, and perceived peer and parental support, when compared to the traditionally delivered G4G program and a control group.

The current intervention evaluation trial of this G4G protocol study presents, as far as the authors are aware, the first three-armed intervention effectiveness study to be conducted in an Irish OYS context, designed to increase participants MVPA, FMS proficiency, and psychological wellbeing among 8–12-year-old girls. This G4G NRCT has provided sufficient evidence regarding the implementation of a multi-component, community-sports based intervention, and can be used as a starting point to inform the development of theory-based interventions targeting PA promotion for pre-adolescent girls in Ireland. Examining the future efficacy of the G4G program, as part of a larger trial would seem prudent in terms of the gathering sustainable, longitudinal evidence.

## Figures and Tables

**Figure 1 ijerph-17-06928-f001:**
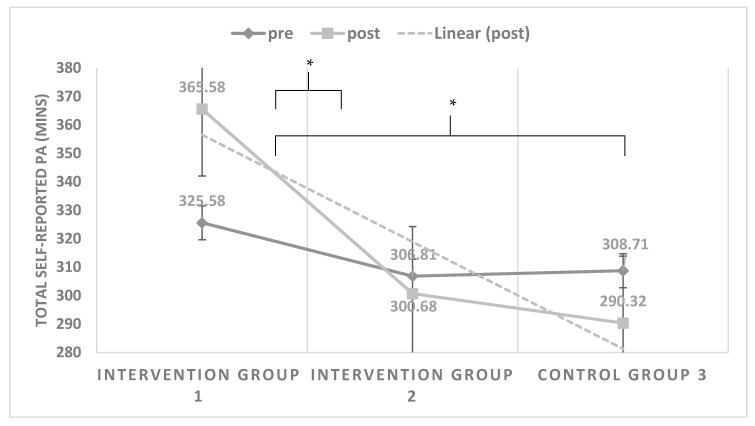
Comparison of the mean self-reported minutes of MVPA over time (pre to post) by group. * Significant difference between group 1 (*p* ≤ 0.003) and group 2 (*p* ≤ 0.002) and between group 1 and 3 (*p* ≤ 0.004).

**Figure 2 ijerph-17-06928-f002:**
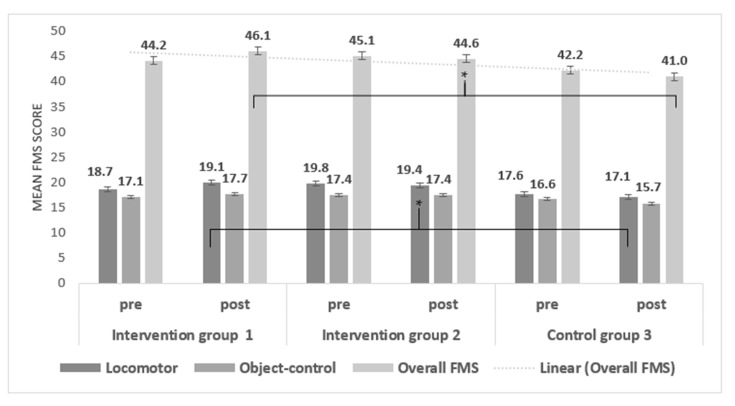
Mean locomotor, object-control, and overall FMS proficiency differences among groups over time (pre to post stages; n = 120). * Significant difference for overall FMS proficiency between group 1 and group 3 (*p* ≤ 0.005). Significant locomotor differences between group 1 and 3 (*p* ≤ 0.000).

**Table 1 ijerph-17-06928-t001:** Overview of the research informed Gaelic4Girls intervention group 1 program components with associated pedagogical considerations.

Part	What	Pedagogical Considerations
**Participant**	- 6 × 10-min rotatory LGF skills stations (i.e., provision of specific time for locomotor and object control skill execution and development) - Sport-specific G4G skill cards (10 in total, e.g., kick, catch, hand pass) of key coaching cue points. - G4G innovative LGF and FMS dance for a duration of 30 min indoor, prior to commencement of G4G outdoor pitch sessions, from weeks 3 to 8 (extra 300 min of FMS specific execution). - Specific “Team Challenges” station to promote fun, friendship and inclusivity.	- TGfU instructional model [53] - SHARP Principles [54] - SAAFE Principles [55]
PA		
FMS		
Psychological		
**Coach**	- Initial 2-h Coach Education Workshop session, prior to the start of the G4G intervention with the lead researcher. - 8 × 1-h Coach Education sessions, specifically with the lead researcher engaging face-to-face with the G4G club coordinators and coaches each week before sessions - G4G Coaching Manual & Resource Pack/Electronic video clips of activities with key coaching points, via booklet and on-demand “WhatsApp” instant messaging service. This instant messaging service was also used for communication correspondence regarding the weekly training session set-up structure. - G4G Coordinator and Coach Reflective Task provided with the opportunity to suggest additions or develop new strategies to engage the G4G participants on a weekly basis.	- SHARP Principles [54] - SAAFE Principles [55] - SDT Interpersonal, Autonomy, Competence and Relatedness
Coach Education		
Workshops		
Resources		
Support		
Structures		
**Parental**	- Technological resources—The lead researcher set up a “WhatsApp” instant messaging and “Facebook” group for all interested parents - On week 4 of the G4G program, the lead researcher designed and delivered a 45 min “G4G Parent/Guardian Workshop” in the community sports setting. - Parent and guardians were encouraged to actively get involved with their daughter through the LGF skills at home, specifically via the G4G Skill Cards.	- Interpersonal (communication/relatedness with other parents - Encourage parents to promote “youth ownership”
Support & Engagement		

Abbreviations: PA = physical activity; FMS = fundamental movement skills; G4G = Gaelic4Girls; TGfU = Teaching Games for Understanding; SHARP = “Stretching whilst moving”, “High repetition of motor skills”, “Accessibility through differentiation”, “Reducing sitting and standing”, “Promoting on pitch physical activity”; SAAFE = “Supportive, Active, Autonomous, Fair, and Enjoyable”.

**Table 2 ijerph-17-06928-t002:** Descriptive statistics [means and standard deviations (M ± SD)] of PA, FMS and psychological measurements, stratified by group and time-period.

Variables	Intervention G1		Intervention G2		Control G3	
	Pre	Post	Pre	Post	Pre	Post
*N*	43	43	44	44	31	31
PA (mins)	325.81 ± 98.76	365.58 ± 64.56	306.81 ± 84.57	300.68 ± 68.32	308.71 ± 85.97	290.32 ± 66.36
LOM	18.67 ± 2.90	19.91 ± 2.70	19.80 ± 2.33	19.37 ± 2.42	18.48 ± 2.71	17.68 ± 2.48
OC	17.12 ± 2.72	17.70 ± 2.58	17.41 ± 2.22	17.43 ± 2.24	16.61 ± 2.22	15.71 ± 1.64
Overall FMS	44.21 ± 5.13	46.07 ± 4.43	45.14 ± 4.07	44.59 ± 4.21	42.21 ± 4.49	41.00 ± 3.79
SE	33.79 ± 4.02	35.60 ± 4.74	32.45 ± 4.38	31.98 ± 4.86	33.13 ± 4.19	31.03 ± 6.16
PSC	118.00 ± 23.24	121.30 ± 23.28	116.12 ± 28.17	115.12 ± 27.64	123.84 ± 26.01	118.10 ± 24.84
Enjoyment	43.00 ± 3.52	45.93 ± 3.48	44.48 ± 3.82	44.30 ± 3.43	42.70 ± 3.99	41.87 ± 3.89
PA Attitudes	48.19 ± 6.37	49.33 ± 5.06	46.07 ± 6.29	45.39 ± 5.71	46.81 ± 5.64	45.42 ± 5.28
Peer/SS	17.79 ± 3.18	19.26 ± 3.32	17.68 ± 3.22	17.00 ± 3.16	18.87 ± 3.90	17.90 ± 3.58
Family S	18.00 ± 3.35	19.49 ±3.53	17.40 ± 3.90	17.71 ± 3.64	18.87 ± 4.36	18.16 ± 4.48

Abbreviations: M = mean; SD = standard deviation; PA =physical activity; LOM = locomotor; OC = object-control; FMS = fundamental movement skills; SE = self-efficacy; PSC: perceived self-confidence; SS = social support.

**Table 3 ijerph-17-06928-t003:** Mean change (MC) differences and significant interaction effects (within time and between-group) for the psychological variables.

Variable	N	G1 (*MC*)	G2 (*MC*)	G3 (*MC*)	Main Interaction Effects	*d*	Post Hoc MC Differences
SE	118	1.81	−0.47	−2.1	*F*(2,115) = 15.18, *p* = 0.000	0.209	G1 & G2 *p* < 0.002 G1 & G3 *p* < 0.003
PSC	114	3.30	−1.00	−5.74	*F*(2,114) = 8.81, *p* = 0.000	0.134	G1 & G3 *p* < 0.002 G2 & G3 *p* < 0.001
PA Enjoyment	118	2.93	−0.18	−0.83	*F*(2,118) = 13.82, *p* = 0.000	0.107	G1 & G3 *p* < 0.003
Attitudes towards PA	115	1.51	−0.48	−0.09	*F*(2,115) = 12.64, *p* = 0.000	0.180	G1 & G2 *p* < 0.003
Peer Social Support	115	1.47	−0.68	−0.97	*F*(2,115) = 37.17, *p* = 0.000	0.393	G1 & G2 *p* < 0.003 G1 & G3 *p* < 0.001
Family Support	113	1.49	0.31	−0.17	*F*(2,115) = 10.16, *p* = 0.000	0.152	G1 & G2 *p* < 0.002

Abbreviations: MC = mean change; G1 = intervention group 1; G2 = intervention group 2; G3 = control group 3; d = Cohen’s d; SE: self-efficacy; PSC: perceived self-confidence, PA: physical activity.

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
