# Peer review of "Gaelic4Girls—The Effectiveness of a 10-Week Multicomponent Community Sports-Based Physical Activity Intervention for 8 to 12-Year-Old Girls"

_ijerph, 2020, doi:10.3390/ijerph17186928_

Round 1
Reviewer 1 Report
The main objective of the study is to assess the effectiveness of a community sports-based intervention for increasing physical activity levels, fundamental movement skill proficiency, and psychological wellbeing of pre-adolescent Irish girls. The authors propose a valid 10-week G4G multi-component community sports-based intervention. In my opinion, they provide relevant and useful information. I think this manuscript could be accepted after minor revision (corrections to minor errors, improvement of Tables, and text editing). I comment on some issues below:
The Introduction is clear, it provides adequate background and includes relevant references.
Materials and Methods
-Line 384: the number of participants is not correct (18)
-Line 406: the number is not correct (One hundred and sixty fifty)
-I would prefer to see written details of psychological measurement instruments instead of Table 1.2.
-Data analysis: Line 485 – Could you provide the exact percentage of participants with incomplete data for a given variable who were excluded from the analysis?
Results
-Please, Tables could be improved. Number them in the correct order, from Table 1 to Table 4. Add abbreviations as notes below the table. Improve the titles, summarized them, remove parentheses.
-Table 1.3. It is difficult to keep reading the data, please add M(SD) instead of M+-SD
The Discussion could be summarized. It is very repetitive of the results. Please, instead of repeating, discuss more, and compare the results with previous related studies.
Author Response
Reviewer 1
Title: ‘Gaelic4Girls’ – The Effectiveness of a 10-Week Multicomponent Community-Sports Based Physical Activity Intervention for 8 to 12-year-old Girls
Manuscript ID: ijerph-911992
Date of Resubmission: 15/09/20
Authors: ORLAGH FARMER 1, KEVIN CAHILL 2, WESLEY O’ BRIEN 1
Affiliations:
1School of Education, Sports Studies and Physical Education, University College Cork, Cork, Ireland.
2 School of Education, Postgraduate Diploma in Special Educational Needs, University College Cork, Cork, Ireland.
Corresponding Author:
Dr. Orlagh Farmer; email: o.farmer@umail.ucc.ie
Authors:
Dr. Kevin Cahill; email: K.Cahill@ucc.ie
Dr. Wesley O’ Brien; email: wesley.obrien@ucc.ie
15/09/20
To whom it may concern (Reviewer 1),
As lead author, I would like to take this opportunity to thank you for taking the time to review our previously submitted research article (Manuscript ID: ijerph-911992) entitled ‘Gaelic4Girls’ – The Effectiveness of a 10-Week Multicomponent Community-Sports Based Physical Activity Intervention for 8 to 12-year-old Girls for consideration in the International Journal of Environmental Research and Public Health.
We have made some minor amendments to this paper, specifically altering the Materials and Methods section, including a written section on the assessment psychological variables and the exact percentage of participants with incomplete data from the analysis. In the Results section, the tables have been improved. Finally, we have made some significant amendments to the Discussion namely;
- Reduced word count and removal of ‘repetitiveness of results’
- Inclusion of more discussion, and comparison of the results with previous related studies (highlighted in yellow).
The document attached is a combination of all four Reviewer’s comments and author responses (including the tracked changes). In order to speed up the revision process, I have color coded the responses to the Reviewers (R1 = blue; R2 = green; R3: yellow; R4: pink). My co-authors and I are hopeful that the updated and revised manuscript will now satisfy the scope and mission of your journal.
I can confirm that neither the manuscript nor any parts of its content are currently under consideration or published in another journal. Furthermore, we declare that there is no conflict of interest related to the research reported in the manuscript. I have read the journal’s mission statement and followed the Instructions for Authors guide while compiling and resubmitting this manuscript for review.
I, Dr. Orlagh Farmer, will be serving as the corresponding author for this manuscript. All of the authors listed in the by-line have agreed to the by-line order and to submission of the manuscript in this form. I have assumed responsibility for keeping my co-authors informed of our progress through the editorial review process; the content of the reviews, and any revisions made.
I want to thank you for allowing my co-authors and I the opportunity to address the previous comments and submit this revised manuscript to the International Journal of Environmental Research and Public Health. I look forward to hearing from you in due course.
Yours Sincerely,
Dr. Orlagh Farmer
School of Education
Sports Studies and Physical Education
University College Cork
Tel: 00353 86 350 9870
Email: o.farmer@umail.ucc.ie

Reviewer 2 Report
I have read the article by Farmer et al. with great interest. The authors compared research informed G4G programme with a traditional one and control circumstances. I would like to congratulate for the nice research design and writing.
Comments:
- Please correct the typo in the title (i.e. “pf” vs. “of”).
- Line 406. Please clarify “One hundred and sixty fifty participants”.
- 1. Could you please provide the p values for the pre-intervention comparison between the groups. I understand that the difference was not significant, but there may be some tendency for differences.
Author Response
Reviewer 2
Title: ‘Gaelic4Girls’ – The Effectiveness of a 10-Week Multicomponent Community-Sports Based Physical Activity Intervention for 8 to 12-year-old Girls
Manuscript ID: ijerph-911992
Date of Resubmission: 15/09/20
Authors: ORLAGH FARMER 1, KEVIN CAHILL 2, WESLEY O’ BRIEN 1
Affiliations:
1School of Education, Sports Studies and Physical Education, University College Cork, Cork, Ireland.
2 School of Education, Postgraduate Diploma in Special Educational Needs, University College Cork, Cork, Ireland.
Corresponding Author:
Dr. Orlagh Farmer; email: o.farmer@umail.ucc.ie
Authors:
Dr. Kevin Cahill; email: K.Cahill@ucc.ie
Dr. Wesley O’ Brien; email: wesley.obrien@ucc.ie
15/09/20
To whom it may concern (Reviewer 2),
As lead author, I would like to take this opportunity to thank you for taking the time to review our previously submitted research article (Manuscript ID: ijerph-911992) entitled ‘Gaelic4Girls’ – The Effectiveness of a 10-Week Multicomponent Community-Sports Based Physical Activity Intervention for 8 to 12-year-old Girls for consideration in the International Journal of Environmental Research and Public Health.
We have made some minor amendments to this paper, specifically amending the typos in the Title and the Materials and Methods section. In the Results section, the p values for pre-intervention have not yet been added. The corresponding author is waiting for the University to grant a new licence renewal number for the statistical software used in order to gain access to the SPSS file. I should be able to gain access over the coming days and will add the p values, as requested.
The document attached is a combination of all four Reviewer’s comments and author responses (including the tracked changes). In order to speed up the revision process, I have color coded the responses to the Reviewers (R1 = blue; R2 = green; R3: yellow; R4: pink). My co-authors and I are hopeful that the updated and revised manuscript will now satisfy the scope and mission of your journal.
I can confirm that neither the manuscript nor any parts of its content are currently under consideration or published in another journal. Furthermore, we declare that there is no conflict of interest related to the research reported in the manuscript. I have read the journal’s mission statement and followed the Instructions for Authors guide while compiling and resubmitting this manuscript for review.
I, Dr. Orlagh Farmer, will be serving as the corresponding author for this manuscript. All of the authors listed in the by-line have agreed to the by-line order and to submission of the manuscript in this form. I have assumed responsibility for keeping my co-authors informed of our progress through the editorial review process; the content of the reviews, and any revisions made.
I want to thank you for allowing my co-authors and I the opportunity to address the previous comments and submit this revised manuscript to the International Journal of Environmental Research and Public Health. I look forward to hearing from you in due course.
Yours Sincerely,
Dr. Orlagh Farmer
School of Education
Sports Studies and Physical Education
University College Cork
Tel: 00353 86 350 9870
Email: o.farmer@umail.ucc.ie

Reviewer 3 Report
Paper is interesting, especially the intervention and the enhancement of an existing women's football club to deliver the intervention for preadolescents in Ireland.
Please see my attached review for further feedback.

Author Response
Reviewer 3
Title: ‘Gaelic4Girls’ – The Effectiveness of a 10-Week Multicomponent Community-Sports Based Physical Activity Intervention for 8 to 12-year-old Girls
Manuscript ID: ijerph-911992
Date of Resubmission: 15/09/20
Authors: ORLAGH FARMER 1, KEVIN CAHILL 2, WESLEY O’ BRIEN 1
Affiliations:
1School of Education, Sports Studies and Physical Education, University College Cork, Cork, Ireland.
2 School of Education, Postgraduate Diploma in Special Educational Needs, University College Cork, Cork, Ireland.
Corresponding Author:
Dr. Orlagh Farmer; email: o.farmer@umail.ucc.ie
Authors:
Dr. Kevin Cahill; email: K.Cahill@ucc.ie
Dr. Wesley O’ Brien; email: wesley.obrien@ucc.ie
15/09/20
To whom it may concern (Reviewer 3),
As lead author, I would like to take this opportunity to thank you for taking the time to review our previously submitted research article (Manuscript ID: ijerph-911992) entitled ‘Gaelic4Girls’ – The Effectiveness of a 10-Week Multicomponent Community-Sports Based Physical Activity Intervention for 8 to 12-year-old Girls for consideration in the International Journal of Environmental Research and Public Health.
We have made significant amendments to this paper, specifically amending the title typo, inclusion of more detail on the importance of the psychological correlates of physical activity in the Introduction section, clarification on the age range of the participants in the Materials and Methods section. To enhance the reader’s understanding of the intervention group 1’s treatment/conditions, we have decided to include a methods supplementary document. In the Results section, the tables and figures have been significantly improved. Finally, we have made some significant amendments to the Discussion namely;
- Reduced word count and removal of ‘repetitiveness of SAAFE and SHARP principles.
- Removal of redundant sentences/restructuring of some sentences.
- Inclusion of more discussion, and comparison of the results with previous related studies (highlighted in yellow).
The document attached is a combination of all four Reviewer’s comments and author responses (including the tracked changes). In order to speed up the revision process, I have color coded the responses to the Reviewers (R1 = blue; R2 = green; R3: yellow; R4: pink). My co-authors and I are hopeful that the updated and revised manuscript will now satisfy the scope and mission of your journal.
I can confirm that neither the manuscript nor any parts of its content are currently under consideration or published in another journal. Furthermore, we declare that there is no conflict of interest related to the research reported in the manuscript. I have read the journal’s mission statement and followed the Instructions for Authors guide while compiling and resubmitting this manuscript for review.
I, Dr. Orlagh Farmer, will be serving as the corresponding author for this manuscript. All of the authors listed in the by-line have agreed to the by-line order and to submission of the manuscript in this form. I have assumed responsibility for keeping my co-authors informed of our progress through the editorial review process; the content of the reviews, and any revisions made.
I want to thank you for allowing my co-authors and I the opportunity to address the previous comments and submit this revised manuscript to the International Journal of Environmental Research and Public Health. I look forward to hearing from you in due course.
Yours Sincerely,
Dr. Orlagh Farmer
School of Education
Sports Studies and Physical Education
University College Cork
Tel: 00353 86 350 9870
Email: o.farmer@umail.ucc.ie

Reviewer 4 Report
The work raised by Farmer et al is very interesting. However, I consider that there are some aspects that need to be improved:
Title
I don't understand the title. Perhaps better to put ‘Gaelic4Girls’ (G4G) at the end of it.
P for of?
Abstract.
Include brief background
Include specific number in the results section.
Rewrite conclusion based on study objectives.
Introduction.
It should be explained why Gaelic4Girls' (G4G) is chosen or what exactly it is since it turns out to be a central aspect of the manuscript and from the title it only appears on line 360.
Methods: List the tables according to appearance: 1, 2, 3, 4, ...
Include table 2 after the Psychological Correlates of PA section.
Lines 504 and 505: I am not sure where these abbreviations come from and if they are wrong to include them there.
Results:
Delete M SD from table 3
Abbreviations in the order of the table.
There were no significant differences (p> 0.05) between treatments in table 3?
Indicate the significant differences in figure 1 so that it can be seen in the figure itself.
Conclusions
Rewrite conclusions or objectives so that both agree.
Author Response
Reviewer 4
Title: ‘Gaelic4Girls’ – The Effectiveness of a 10-Week Multicomponent Community-Sports Based Physical Activity Intervention for 8 to 12-year-old Girls
Manuscript ID: ijerph-911992
Date of Resubmission: 15/09/20
Authors: ORLAGH FARMER 1, KEVIN CAHILL 2, WESLEY O’ BRIEN 1
Affiliations:
1School of Education, Sports Studies and Physical Education, University College Cork, Cork, Ireland.
2 School of Education, Postgraduate Diploma in Special Educational Needs, University College Cork, Cork, Ireland.
Corresponding Author:
Dr. Orlagh Farmer; email: o.farmer@umail.ucc.ie
Authors:
Dr. Kevin Cahill; email: K.Cahill@ucc.ie
Dr. Wesley O’ Brien; email: wesley.obrien@ucc.ie
15/09/20
To whom it may concern (Reviewer 4),
As lead author, I would like to take this opportunity to thank you for taking the time to review our previously submitted research article (Manuscript ID: ijerph-911992) entitled ‘Gaelic4Girls’ – The Effectiveness of a 10-Week Multicomponent Community-Sports Based Physical Activity Intervention for 8 to 12-year-old Girls for consideration in the International Journal of Environmental Research and Public Health.
We have made some minor amendments to this paper, specifically altering the Abstract, improving the tables and figures in the Materials and Methods section and the Results section. Finally, the Conclusion is amended so that the study aim’s and objectives agree with the conclusion.
The document attached is a combination of all four Reviewer’s comments and author responses (including the tracked changes). In order to speed up the revision process, I have color coded the responses to the Reviewers (R1 = blue; R2 = green; R3: yellow; R4: pink). My co-authors and I are hopeful that the updated and revised manuscript will now satisfy the scope and mission of your journal.
I can confirm that neither the manuscript nor any parts of its content are currently under consideration or published in another journal. Furthermore, we declare that there is no conflict of interest related to the research reported in the manuscript. I have read the journal’s mission statement and followed the Instructions for Authors guide while compiling and resubmitting this manuscript for review.
I, Dr. Orlagh Farmer, will be serving as the corresponding author for this manuscript. All of the authors listed in the by-line have agreed to the by-line order and to submission of the manuscript in this form. I have assumed responsibility for keeping my co-authors informed of our progress through the editorial review process; the content of the reviews, and any revisions made.
I want to thank you for allowing my co-authors and I the opportunity to address the previous comments and submit this revised manuscript to the International Journal of Environmental Research and Public Health. I look forward to hearing from you in due course.
Yours Sincerely,
Dr. Orlagh Farmer
School of Education
Sports Studies and Physical Education
University College Cork
Tel: 00353 86 350 9870
Email: o.farmer@umail.ucc.ie

Round 2
Reviewer 4 Report
The authors have greatly improved the manuscript. If the editor considers, it can be accepted in the present form.